# Pyrotinib after trastuzumab-based adjuvant therapy in patients with HER2-positive breast cancer (PERSIST): A multicenter phase II trial

Feilin Cao[1]*[†], Zhaosheng Ma[1†], Zenggui Wu[1†], Weizhu Wu[2], Ouchen Wang[3], Binbin Cui[1‡], Xiaotao Zhu[4‡], Jing Hao[5§], Xiaochun Ji[2§], Zhanwen Li[6§], Deyou Tao[1], Qingjing Feng[7#], Wei Lin[8#], Dongbo Shi[9#], Jingde Shu[8#], Jichun Zhou[10], Shifen Huang[1]

[1]Department of Thyroid and Breast Surgery, Taizhou Hospital of Zhejiang Province, Wenzhou Medical University, Taizhou, China; [2]Department of Thyroid and Breast Surgery, Ningbo Medical Center Lihuili Hospital, Ningbo, China; [3]Department of Breast Surgery, The First Affiliated Hospital of Wenzhou Medical University, Wenzhou, China; [4]Department of Thyroid and Breast Surgery, Jinhua Municipal Central Hospital, Jinhua, China; [5]Department of Thyroid and Breast Surgery, Yiwu Central Hospital, Jinhua, China; [6]Department of Breast Surgery, Ningbo Women and Children's Hospital, Ningbo, China; [7]Department of Breast Surgery, Yiwu Maternity and Children Hospital, Jinhua, China; [8]Department of Surgical Oncology, Quzhou People's Hospital, Quzhou, China; [9]Department of Breast Surgery, The First Affiliated Hospital of Ningbo University, Ningbo, China; [10]Department of Surgical Oncology, Sir Run Run Shaw Hospital, Zhejiang University School of Medicine, Hangzhou, China

*For correspondence:
drcfl@126.com

[†]These authors contributed equally to this work
[‡]These authors also contributed equally to this work
[§]These authors also contributed equally to this work
[#]These authors also contributed equally to this work

Competing interest: The authors declare that no competing interests exist.

## eLife Assessment

The study by Ma et al. provides **fundamental** findings and **compelling** evidence that Pyrotinib after trastuzumab-based adjuvant therapy in patients with HER2-positive breast cancer (PERSIST): A multi-center phase II trial. The findings enhance the understanding of HER2-positive breast cancer. The claims are fully supported by the types of experiments that were performed.

## Abstract

**Background:** Approximately one-third of patients with HER2-positive breast cancer experienced recurrence within 10 years after receiving 1 year of adjuvant trastuzumab. The ExteNET study showed that 1 year of extended adjuvant neratinib after trastuzumab-based adjuvant therapy could reduce invasive disease-free survival (iDFS) events compared with placebo. This study investigated the efficacy and safety of pyrotinib, an irreversible pan-HER receptor tyrosine kinase inhibitor, after trastuzumab-based adjuvant therapy in patients with high-risk, HER2-positive early or locally advanced breast cancer.

**Methods:** This multicenter phase II trial was conducted at 23 centers in China. After enrollment, patients received 1 year of extended adjuvant pyrotinib (400 mg/day), which should be initiated within 6 months after the completion of 1-year adjuvant therapy (trastuzumab alone or plus pertuzumab). The primary endpoint was 2-year iDFS rate.

**Results:** Between January 2019 and February 2022, 141 eligible women were enrolled and treated. As of October 10, 2022, the median follow-up was 24 (interquartile range, 18.0–34.0) months. The 2-year iDFS rate was 94.59% (95% confidence interval [CI]: 88.97–97.38) in all patients, 94.90% (95% CI: 86.97–98.06) in patients who completed 1-year treatment, 90.32% (95% CI: 72.93–96.77) in patients who completed only 6-month treatment, 96.74% (95% CI: 87.57–99.18) in the hormone receptor (HR)-positive subgroup, 92.77% (95% CI: 83.48–96.93) in the HR-negative subgroup, 96.88% (95% CI: 79.82–99.55) in the lymph node-negative subgroup, 93.85% (95% CI: 86.81–97.20) in the lymph node-positive subgroup, 97.30% (95% CI: 82.32–99.61) in patients with adjuvant trastuzumab plus pertuzumab, and 93.48% (95% CI: 86.06–97.02) in patients with adjuvant trastuzumab. The most common adverse events were diarrhea (79.4%), fatigue (36.9%), lymphocyte count decreased (36.9%), nausea (33.3%), and hand-foot syndrome (33.3%).

**Conclusions:** Extended adjuvant pyrotinib administered after trastuzumab-based adjuvant therapy showed promising efficacy in patients with high-risk HER2-positive breast cancer. The follow-up is ongoing to determine the long-term benefit.

**Funding:** No external funding was received for this work.

**Clinical trial number:** ClinicalTrials.gov: NCT05880927

## Introduction

Worldwide, breast cancer is the most commonly diagnosed malignancy in women. Approximately 15–20% of breast cancer cases are characterized by the overexpression of human epidermal growth factor receptor 2 (HER2) (*Siegel et al., 2014*; *Loibl and Gianni, 2017*). Over the past 20 years, with the emergence of HER2-targeted agents, it has been proved that the addition of 1-year trastuzumab to adjuvant chemotherapy significantly reduces the recurrence rate and prolongs survival in patients with HER2-positive breast cancer (*Perez et al., 2014*; *Cameron et al., 2017*; *Wolff et al., 2013*). However, the phase III HERA study found that, despite receiving 1 year of adjuvant trastuzumab, 31% of patients with HER2-positive breast cancer experienced recurrence within 10 years (*Cameron et al., 2017*). In addition, the phase III APHINITY study reported a 12% recurrence rate in HER2-positive breast cancer patients with node-positive disease after receiving 1 year of adjuvant trastuzumab, pertuzumab, and chemotherapy with a median follow-up of 74 months (*Piccart et al., 2021*). Consequently, novel treatment approaches are warranted to extend the time to recurrence.

Currently, the efficacy of extended adjuvant therapy has only been investigated in the phase III ExteNET study. The results showed that 1 year of neratinib after trastuzumab-based adjuvant therapy significantly improved invasive disease-free survival (iDFS) in women with HER2-positive breast cancer. However, the iDFS benefit did not translate into overall survival benefit in the intention-to-treat population after 8-year follow-up, except for patients with hormone receptor (HR)-positive disease (*Holmes et al., 2023*; *Lüftner et al., 2021*; *Martin et al., 2017*).

Pyrotinib is an oral irreversible pan-HER receptor tyrosine kinase inhibitor (TKI), targeting epidermal growth factor receptor, HER2, and HER4 (*Ma et al., 2019*; *Guan et al., 2023b*; *Xu et al., 2021*). For patients with HER2-positive metastatic breast cancer, improved survival outcomes were shown in the pyrotinib group when compared with the control group in the phase III PHOEBE and PHILA studies (*Ma et al., 2023*; *Wu et al., 2022*). Additionally, results from the phase III PHEDRA trial demonstrated clinical benefits of pyrotinib plus trastuzumab and docetaxel for patients with HER2-positive early or locally advanced breast cancer in the neoadjuvant setting (*Chan et al., 2016*).

In the present PERSIST study, our objective was to assess the efficacy and safety of administering extended adjuvant pyrotinib treatment following trastuzumab-based adjuvant therapy in patients with high-risk, HER2-positive early or locally advanced breast cancer. This report presents the efficacy and safety results with a median follow-up of 2 years, while longer follow-up is still ongoing.

## Methods
### Study design and participants

This multicenter, single-arm, investigator-initiated, phase II exploratory study was conducted at 23 sites in China. The study protocol was approved by the institutional ethics committee of Taizhou Hospital of Zhejiang Province and all other participating centers, and the trial was performed in accordance with

the 2013 Declaration of Helsinki. This trial was registered at https://clinicaltrials.gov/ (NCT05880927). Written informed consent was obtained from all patients before study enrollment.

Participants were eligible if they were women aged 18–75 years with high-risk, HER2-positive (immunohistochemistry 3+, or 2+ with gene amplification by fluorescence in situ hybridization) early or locally advanced invasive breast cancer; had previously completed 1 year of adjuvant therapy with either trastuzumab or trastuzumab plus pertuzumab within 6 months before enrollment; had an Eastern Cooperative Oncology Group performance status of 0 or 1; had known HR status; and had adequate bone marrow, hepatic, renal, and cardiac functions.

High-risk patients were defined as those meeting one of the following criteria: N stage ≥1; T stage ≥2; absence of pathological complete response (pCR) after neoadjuvant therapy; presence of pCR after neoadjuvant therapy but with tumor size ≥5 cm or N stage ≥2; tumor size <2 cm but with high Ki-67 level or histologic grade 3. Clinical and pathological stage was assessed according to the eighth edition of American Joint Committee on Cancer tumor-node-metastasis staging system. The expression of Ki-67 was categorized as low (<20%) and high (≥20%) based on the ratio of positive cells to all tumor cells in 10 high-power fields.

The main exclusion criteria were bilateral breast cancer, inflammatory breast cancer, previous participation in other clinical trials involving anticancer therapy, allergies to the drug components in this protocol, history of immunodeficiency diseases, inability to swallow, and pregnancy or lactation.

### Procedure

Pyrotinib 400 mg was administrated orally once daily within 6 months of completing trastuzumab-based adjuvant therapy. Concurrent adjuvant endocrine therapy for HR-positive disease was recommended.

Treatment duration was 12 months unless intolerable adverse events, disease recurrence, death, or consent withdrawal occurred.

### Outcomes and assessment

The primary endpoint was iDFS rate at 2 years. IDFS was defined as time from the initiation of pyrotinib treatment to invasive ipsilateral tumor recurrence, invasive contralateral breast cancer, local or regional invasive recurrence, distant recurrence, or death from any cause. The secondary endpoint was safety profile, graded according to the National Cancer Institute Common Terminology Criteria for Adverse Events, version 4.0.

Patients were followed every 3 months for the first 2 years. The follow-up assessments included physical examination, Eastern Cooperative Oncology Group performance status, complete blood count, blood chemistry analysis, 12-lead electro-cardiogram, recording of adverse events, concomitant medications, and ultrasound. Computed tomography and bone scans may be conducted if clinically indicated.

### Statistical analysis

Based on the previous ExteNET trial (*Burstein et al., 2021*), the 2-year iDFS rate was assumed as 91% in the present study. To restrain the width of 2-sided 95% confidence interval (CI) to no more than 0.1, 126 patients were required. Considering a dropout rate of 10%, a total of 140 participants were recruited.

Continuous data were presented as median (range or interquartile range), while categorical data were presented as frequency (percentage). IDFS was estimated using the Kaplan–Meier method, and the corresponding 95% CI was estimated using the Brookmeyer–Crowley method. Efficacy analysis was performed in the full analysis set, which included all patients who received at least one dose of study drug. Safety analysis was performed in the safety set, defined as all patients who received at least one dose of the study drug and underwent at least one safety assessment.

## Results

### Baseline characteristics

Between January 2019 and February 2022, a total of 141 eligible women were enrolled in the study and received pyrotinib. All these patients were included in the full analysis set and safety set. The median age was 50 years (range, 25–72). Of the 141 patients, 107 (75.9%) patients had node-positive

**Table 1.** Baseline characteristics of patients in the full analysis set.

| Variables | All (*n* = 141) |
|---|---|
| Age (years), median (range) | 50 (25–72) |
| **T stage, *n* (%)** | |
| T1 | 43 (30.5%) |
| T2 | 83 (58.9%) |
| T3 | 11 (7.8%) |
| T4 | 2 (1.4%) |
| Unable to determine | 2 (1.4%) |
| **Nodal status, *n* (%)** | |
| Negative | 34 (24.1%) |
| 1–3 positive nodes | 49 (34.8%) |
| >3 positive nodes | 58 (41.1%) |
| **Hormone receptor status, *n* (%)** | |
| Negative (ER and PR negative) | 77 (54.6%) |
| Positive (ER positive, PR positive, or both) | 64 (45.4%) |
| **Ki-67, *n* (%)** | |
| ≥20% | 119 (84.4%) |
| <20% | 22 (15.6%) |
| **Previous neoadjuvant therapy, *n* (%)** | 26 (17.7%) |
| **Previous adjuvant therapy, *n* (%)** | |
| Trastuzumab | 92 (65.2%) |
| Trastuzumab plus pertuzumab | 49 (34.8%) |

Tumors were assessed as being ER or PR positive with a threshold of 1%.
ER = estrogen receptor. PR = progesterone receptor.

disease, and 64 (45.4%) had HR-positive disease. Detailed baseline characteristics are summarized in *Table 1*.

In terms of treatment duration, 11 patients discontinued their treatment due to adverse events, while 38 patients had a treatment duration of less than or equal to 6 months due to the COVID-19 pandemic or personal reasons. A total of 92 and 31 patients completed 1-year and 6-month treatment of extended adjuvant pyrotinib, respectively (*Figure 1*).

## Efficacy

As of October 10, 2022, the median follow-up duration was 24 (interquartile range, 18.0–34.0) months. Of the 141 patients, seven patients experienced iDFS events, including one local recurrence, three brain metastases, two lung metastases, and one death due to distant metastasis. The 2-year iDFS rate was 94.59% (95% CI: 88.97–97.38; *Figure 2*).

Subgroup analyses showed that the 2-year iDFS rate was 94.90% (95% CI: 86.97–98.06) in the 1-year treatment subgroup and 90.32% (95% CI: 72.93–96.77) in the 6-month treatment subgroup (*Figure 3A*). Concerning HR status, the 2-year iDFS rate was 96.74% (95% CI: 87.57–99.18) in patients with HR-positive breast cancer and 92.77% (95% CI: 83.48–96.93) in patients with HR-negative breast cancer (*Figure 3B*). In terms of lymph node status, the 2-year iDFS rate was 96.88% (95% CI: 79.82–99.55) in patients with lymph node-negative breast cancer and 93.85% (95% CI: 86.81–97.20) in patients with lymph node-positive breast cancer (*Figure 3C*). Regarding trastuzumab-based adjuvant regimen, the 2-year iDFS rate was 97.30% (95% CI: 82.32–99.61) in patients who received adjuvant

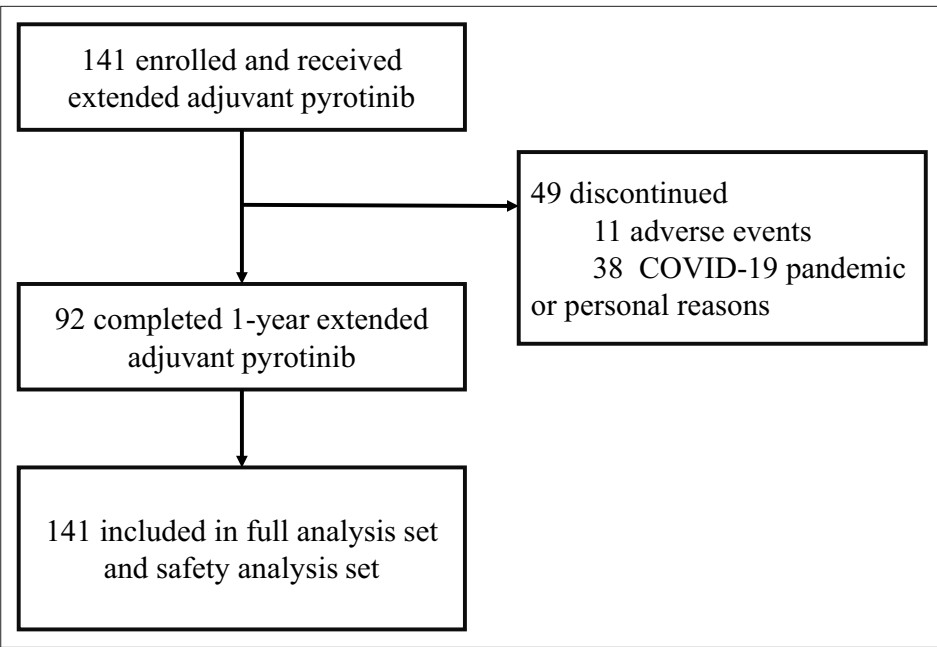

**Figure 1.** Study flowchart.

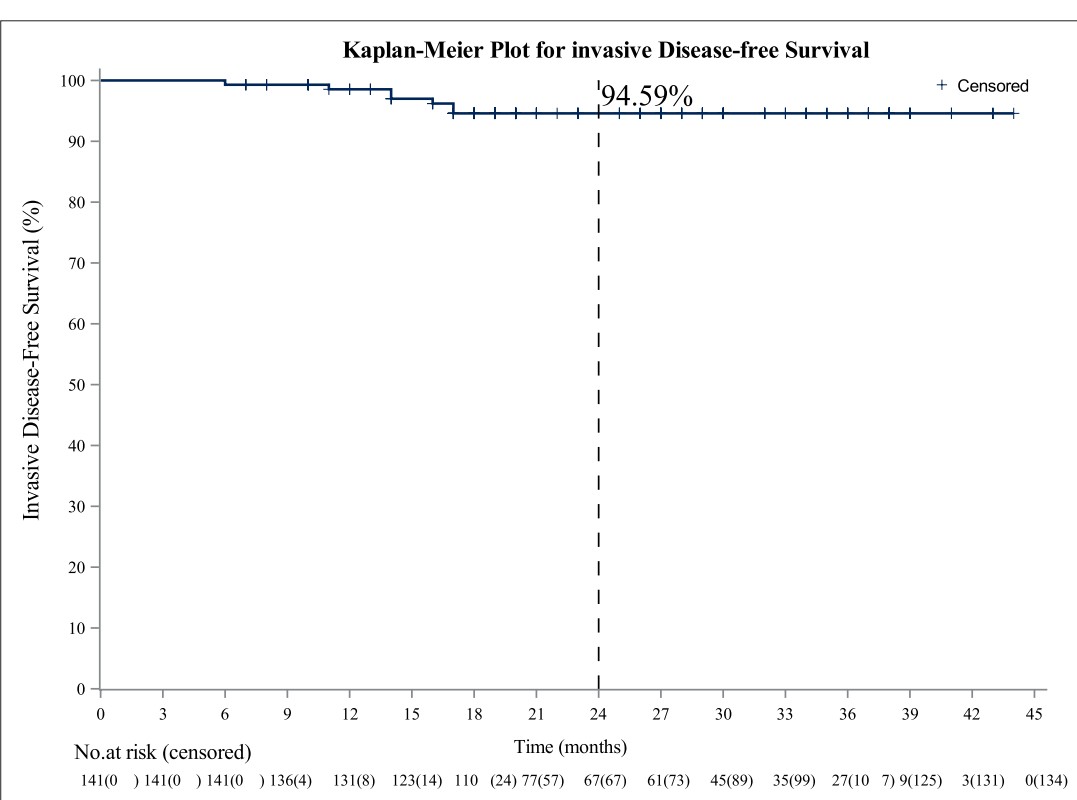

**Figure 2.** Kaplan–Meier estimates of invasive disease-free survival in the full analysis set.

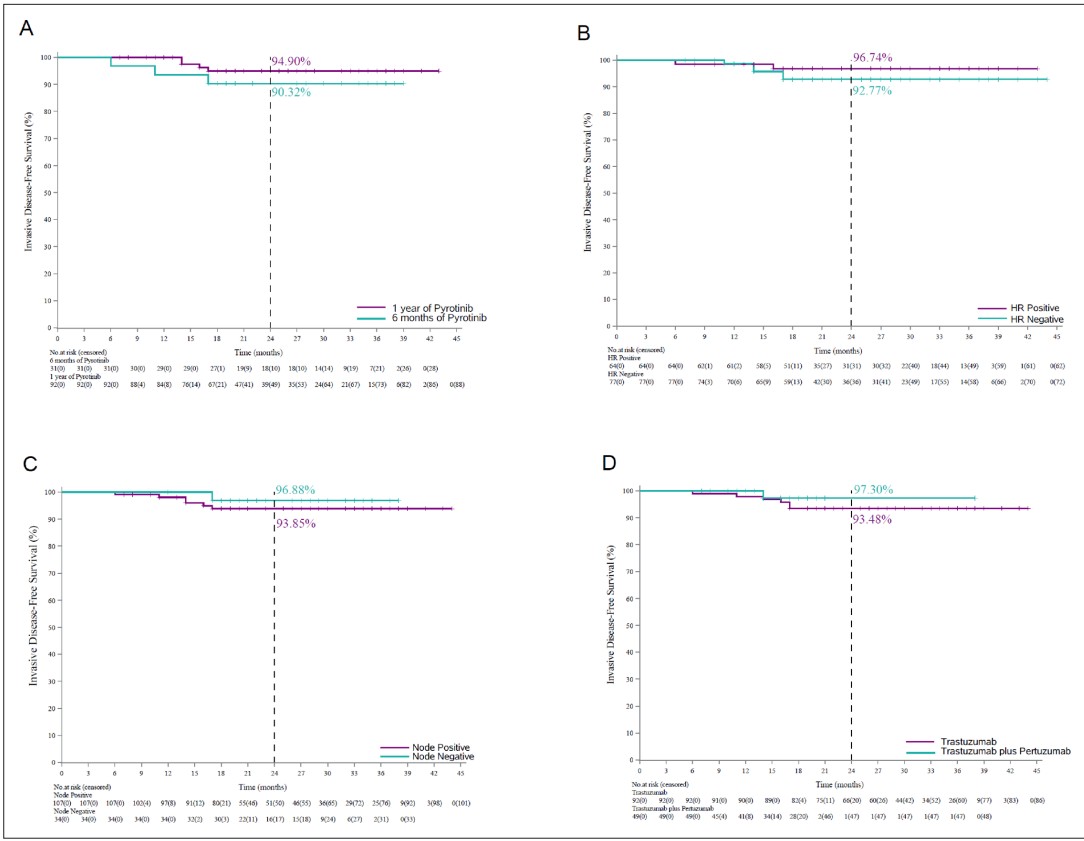

**Figure 3.** Invasive disease-free survival in different subgroups. Kaplan–Meier curves for invasive disease-free survival in (**A**) patients who received 6-month (*n* = 31) and 1-year pyrotinib treatment (*n* = 92), (**B**) patients with hormone receptor-positive breast cancer (*n* = 64) and hormone receptor-negative breast cancer (*n* = 77), (**C**) patients with lymph node-positive breast cancer (*n* = 107) and patients with lymph node-negative breast cancer (*n* = 34), and (**D**) patients who received adjuvant therapy with trastuzumab (*n* = 92) and trastuzumab plus pertuzumab (*n* = 49).

trastuzumab plus pertuzumab, and 93.48% (95% CI: 86.06–97.02) in patients who received adjuvant trastuzumab (*Figure 3D*).

## Safety

The most frequently reported treatment-emergent adverse event was diarrhea (112 [79.4%]), followed by fatigue (52 [36.9%]), lymphocyte count decreased (52 [36.9%]), nausea (47 [33.3%]), and hand-foot syndrome (47 [33.3%]). Forty-three patients (30.5%) experienced grade 3 diarrhea, while no grade 4 or higher adverse events was reported (*Table 2*). Treatment discontinuation occurred in 11 (7.8%) patients, due to grade 3 diarrhea (*n* = 8), grade 3 hand-foot syndrome (*n* = 1), and intolerance (*n* = 2). In addition, 16 patients had dose reduction because of adverse events.

## Discussion

To the best of our knowledge, this multicenter phase II study is the first prospective study to explore the efficacy and safety of extended adjuvant pyrotinib after trastuzumab-based adjuvant therapy in patients with high-risk HER2-positive breast cancer. Furthermore, while the ExteNET study only included patients who received neoadjuvant or adjuvant trastuzumab, over one-third of the patients in our study had received dual anti-HER2 blockade with trastuzumab and pertuzumab, making it more clinically applicable to current clinical practice (*Bevers et al., 2023*). In China, pertuzumab was approved for the treatment of HER2-positive breast cancer in the adjuvant setting in 2018, and has been covered by medical insurance since 2020. The PERSIST study started in 2019, thus the proportion

**Table 2.** Treatment-emergent adverse events occurring in at least 10% of the patients.

| Events, n (%) | All (n = 141) | |
| --- | --- | --- |
| | Any grade | Grade 3 |
| Diarrhea | 112 (79.4%) | 43 (30.5%) |
| Fatigue | 52 (36.9%) | 0 |
| Lymphocyte count decreased | 52 (36.9%) | 0 |
| Nausea | 47 (33.3%) | 0 |
| Hand-foot syndrome | 47 (33.3%) | 1 (0.7%) |
| Hyperuricemia | 33 (23.4%) | 0 |
| White blood cell count decreased | 26 (18.4%) | 0 |
| Dizziness | 26 (18.4%) | 0 |
| Anemia | 24 (17.0%) | 0 |
| Vomiting | 22 (15.6%) | 0 |
| Headache | 19 (13.5%) | 0 |
| Creatinine increased | 17 (12.1%) | 0 |
| Neutrophil count decreased | 16 (11.3%) | 0 |

of eligible patients who received trastuzumab and pertuzumab in our study was relatively high. In addition, we conducted subgroup analyses to explore the differences in the efficacy of extended adjuvant pyrotinib across different prior adjuvant regimens, including trastuzumab plus pertuzumab or trastuzumab alone, and different treatment durations.

Currently, only neratinib has been approved by Food and Drug Administration in the extended adjuvant setting for patients with HER2-positive early or locally advanced breast cancer. In the 2023 National Comprehensive Cancer Network guidelines, it has been recommended for the treatment of high-risk, HR-positive, HER2-positive breast cancer (*Nahta and O'Regan, 2012*). In the ExteNET study, 1-year neratinib treatment after adjuvant trastuzumab significantly improved iDFS in patients with HER2-positive breast cancer, with a 2-year iDFS rate of 93.9% in the neratinib group and 91.6% in the placebo group (*Martin et al., 2017*; *Burstein et al., 2021*). In our study, the 2-year iDFS rate was 94.59%, comparable to that in the ExteNET study.

In terms of subgroup analyses, the HR-positive group seemed to show a numerically higher 2-year iDFS rate than the HR-negative group (96.74% vs 92.77%). Consistently, similar results were observed in the ExteNET study. The underlying mechanism may involve cross-talk between HR and HER2 pathways (*Pegram et al., 2023*; *Sudhan et al., 2019*). Preclinical studies have shown that in estrogen receptor+/HER2+ cell lines, neratinib treatment also suppresses the activity of estrogen reporter (*Hu et al., 2023*). In the present study, adjuvant endocrine therapy was recommended for patients with HR-positive disease when treated with pyrotinib, which may generate synergistic effects (*Zhang et al., 2022*; *Pivot et al., 2019*).

Regarding duration of treatment, the results showed that the 2-year iDFS rate was numerically higher with 1-year pyrotinib treatment (94.90%) versus 6-month pyrotinib treatment (90.32%). In the adjuvant setting, a phase III PHARE trial compared 1-year adjuvant trastuzumab with 6-month treatment, and the non-inferiority was not achieved for disease-free survival. Therefore, 1-year adjuvant trastuzumab remained the standard treatment (*von Minckwitz et al., 2017*). Further studies are warranted to find the optimal duration of extended adjuvant pyrotinib.

Adjuvant treatment regimens before extended adjuvant treatment may also be associated with iDFS. In the present study, the 2-year iDFS rate was 93.48% and 97.30% in the trastuzumab subgroup and trastuzumab plus pertuzumab subgroup, respectively. The APHINITY study demonstrated a significantly improved iDFS with adjuvant trastuzumab plus pertuzumab than with adjuvant trastuzumab alone (*von Minckwitz et al., 2019*). Therefore, the iDFS benefits may last and affect the results in the extended adjuvant period.

None of the patients in the PERSIST study received trastuzumab emtansine (T-DM1), which is now a standard of care for HER2-positive patients with residual cancer after neoadjuvant therapy based on the phase III KATHERINE trial (*Liang et al., 2024*). Since T-DM1 was approved in China in 2020 and was not covered by medical insurance until 2023, it has not been widely used in clinical practice. Recently, a real-world study showed that patients with metastatic breast cancer could benefit from pyrotinib-based therapy after T-DM1 treatment (*Shyam Sunder et al., 2023*). Therefore, patients with high-risk breast cancer may also benefit from extended adjuvant pyrotinib after being treated with T-DM1 in the adjuvant setting, which needs further investigation.

Regarding adverse events, diarrhea was the major safety concern in the present study, leading to treatment discontinuation in eight patients. This adverse event as well as vomiting and nausea are TKI-related common adverse events. The underlying mechanisms of these adverse events are not clear yet, and the mainstream hypothesis is that the growth and healing of epithelial cells in gastrointestinal tract are inhibited by TKIs (*Secombe et al., 2020*; *Guan et al., 2023a*). In the present study, 43 (30.5%) patients experienced grade 3 diarrhea, which were consistent with previous studies of pyrotinib (*Guan et al., 2023b*; *Fang et al., 2022*). Noteworthy, primary prophylaxis was not mandatory in this PERSIST trial. Currently, loperamide hydrochloride has been found to significantly alleviate symptoms of diarrhea and reduce diarrhea-related treatment discontinuation. It has been recommended for patients treated with pyrotinib (*Zheng et al., 2023*). In the PANDORA trial of pyrotinib plus docetaxel, the incidence of diarrhea was significantly lower in the loperamide prophylaxis group (8.9%) compared with the non-prophylaxis group (38.2%) (*Barcenas et al., 2020*). In the CONTROL trial, all cohorts that received loperamide prophylaxis and the dose escalation cohort of neratinib exhibited lower grade 3 diarrhea rates and fewer instances of diarrhea-related treatment discontinuation compared with ExteNET (*Barcenas et al., 2020*). The above two studies indicated that pyrotinib-related diarrhea could be controlled with prophylaxis or dose modification.

There are several limitations in the present study. First, our study was a single-arm study without control group. Second, patients who received T-DM1 therapy in the adjuvant setting were not enrolled. Third, 27.0% of patients did not complete 1-year extended adjuvant pyrotinib due to COVID-19 pandemic or personal reasons. Finally, the follow-up time was short. A phase III clinical trial (NCT03980054) of extended adjuvant pyrotinib in patients with high-risk HER2-positive breast cancer is ongoing, which will further demonstrate the efficacy of extended adjuvant TKI and bring new hope for patients with high-risk HER2-positive breast cancer.

## Conclusions

Our study suggests the potential of extended adjuvant pyrotinib in patients with high-risk HER2-positive early or locally advanced breast cancer after trastuzumab-based adjuvant therapy. The follow-up is ongoing to determine the long-term benefit of extended adjuvant pyrotinib.

## Clinical trial registration number

This trial was registered at https://clinicaltrials.gov/ (NCT05880927).

## Acknowledgements

We thank all the patients who participated in this trial and their families, as well as the investigators and staff. We thank Yan Luo (Department of Medical Affairs, Jiangsu Hengrui Pharmaceuticals Co, Ltd) for data interpretation and Yunning Yang (Department of Medical Affairs, Jiangsu Hengrui Pharmaceuticals Co, Ltd) for medical writing assistance according to Good Publication Practice Guidelines.

## Additional information

### Funding
No external funding was received for this work.

## Author contributions
Feilin Cao, Conceptualization, Supervision, Writing – review and editing; Zhaosheng Ma, Conceptualization, Investigation, Writing – original draft; Zenggui Wu, Investigation, Methodology, Writing – original draft; Weizhu Wu, Software, Formal analysis, Methodology, Writing – original draft; Ouchen Wang, Resources, Methodology, Writing – review and editing; Binbin Cui, Software, Investigation, Project administration; Xiaotao Zhu, Resources, Investigation, Writing – original draft; Jing Hao, Investigation, Writing – original draft, Writing – review and editing; Xiaochun Ji, Conceptualization, Supervision; Zhanwen Li, Resources, Methodology, Project administration; Deyou Tao, Software, Validation; Qingjing Feng, Supervision, Validation, Methodology, Writing – original draft; Wei Lin, Supervision, Methodology, Writing – original draft, Writing – review and editing; Dongbo Shi, Supervision, Validation, Visualization, Writing – original draft, Writing – review and editing; Jingde Shu, Software, Writing – review and editing; Jichun Zhou, Supervision, Validation, Methodology, Writing – review and editing; Shifen Huang, Resources, Writing – review and editing

## Author ORCIDs
Feilin Cao (iD) https://orcid.org/0009-0002-5083-1462

## Ethics
registration NCT05880927.
The study protocol was approved by the institutional ethics committee of Taizhou Hospital of Zhejiang Province and all other participating centers.

Reviewer #1 (Public review): https://doi.org/10.7554/eLife.101724.3.sa1
Author response https://doi.org/10.7554/eLife.101724.3.sa2

# Additional files

## Supplementary files
MDAR checklist

Source data 1. Processed raw data (with all private patient information removed).

## Data availability
All data used in the manuscript (with all private patient information removed) is included with the article as Source Data File 1. This file contains all the data used for the tables and for generating the graphs and charts.

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
