## [Editor Report · eLife Assessment]

The study by Ma et al. provides **fundamental** findings and **compelling** evidence that Pyrotinib after trastuzumab-based adjuvant therapy in patients with HER2-positive breast cancer (PERSIST): A multicenter phase II trial. The findings enhance the understanding of HER2-positive breast cancer. The claims are fully supported by the types of experiments that were performed.

---

## [Referee Report · Reviewer #1 (Public review)]

Summary:

This study introduces a novel therapeutic strategy for patients with high-risk HER2-positive breast cancer and demonstrates that the incorporation of pyrotinib into adjuvant trastuzumab therapy can improve invasive disease-free survival.

Strengths:

The study features robust logic and high-quality data. Data from 141 patients across 23 centers were analyzed, thereby effectively mitigating regional biases and endowing the research findings with high applicability.

Weaknesses:

(1) Introduction and Discussion: Update the literature regarding the efficacy of pyrotinib combined with trastuzumab in treating HER2-positive advanced breast cancer.

(2) Did all the data have a normal distribution? Expand the description of statistical analysis.

(3) The novelty and innovative potential of your manuscript compared to the published literature should be described in more detail in the abstract and discussion section.

(4) Figure legend should provide a bit more detail about what readers should focus on.

(5) P-values should be clarified for the analysis.

(6) The order (A, B, and C) in Figure 3 should be labeled in the upper left corner of the Figure.

Comments on revisions:

The authors responded well to my questions.

---

## [Author Response]

The following is the authors’ response to the original reviews.

**Public Reviews:**

**Reviewer #1 (Public review):**
Summary:This study introduces a novel therapeutic strategy for patients with high-risk HER2-positive breast cancer and demonstrates that the incorporation of pyrotinib into adjuvant trastuzumab therapy can improve invasive disease-free survival.Strengths:The study features robust logic and high-quality data. Data from 141 patients across 23 centers were analyzed, thereby effectively mitigating regional biases and endowing the research findings with high applicability.Weaknesses:(1) Introduction and Discussion: Update the literature regarding the efficacy of pyrotinib combined with trastuzumab in treating HER2-positive advanced breast cancer.

Thank you for this helpful suggestion. The literature regarding the efficacy of pyrotinib combined with trastuzumab in treating HER2-positive advanced breast cancer referenced in our manuscript was the PHILA study, but we mistakenly cited its corrections (reference 14). We revised this reference as suggested.

Changes in the text: Page 6, line 347-353.

(2) Did all the data have a normal distribution? Expand the description of statistical analysis.

As the sample size increases, the sampling distribution of the mean follows a normal distribution even when the underlying distribution of the original variable is non-normal, allowing the use of a normal distribution to calculate their confidence interval. We believe it is unnecessary to specifically describe whether the data followed a normal distribution in this study. Therefore, we did not revise the statistical section.

(3) The novelty and innovative potential of your manuscript compared to the published literature should be described in more detail in the abstract and discussion section.

Thank you for your suggestion. The word count for abstracts recommended by eLife is around 250 words. Therefore, we did not compare the present study with published literature in detail in the abstract, as this might exceed the recommended word limit. We revised the discussion section to provide a more detailed comparison between published literature and our study, and to analyze the novelty of our findings accordingly.

Changes in the text: Page 11, line 177-180.

(4) Figure legend should provide a bit more detail about what readers should focus on.

Thank you for this suggestion. We did not revise the figure legend of Figure 1, as it provides a common description. For the figure legend of Figure 2, we added the method used to estimate the invasive disease-free survival curve. For the figure legend of Figure 3, we added more details regarding methods and numbers of patients in different subgroups.

Changes in the text: Page 7, line 463-472.

(5) P-values should be clarified for the analysis.

Thank you for this comment. All subgroup analyses were post-hoc and lacked predefined hypotheses. Kaplan-Meier curves were used to present the subgroup results with the aim of performing descriptive statistics rather than inferential statistics. Therefore, we did not calculate their p-values.

(6) The order (A, B, and C) in Figure 3 should be labeled in the upper left corner of the Figure.

Thanks for this comment. We revised Figure 3 accordingly.

Changes in the text: Figure 3.

**Reviewer #2 (Public review):**
In this manuscript, Cao et al. evaluated the efficacy and safety of 12 months pyrotinib after trastuzumab-based adjuvant therapy in patients with high-risk, HER2-positive early or locally advanced breast cancer. Notably, the 2-year iDFS rate reached 94.59% (95% CI: 88.97-97.38) in all patients, and 94.90% (95% CI: 86.97-98.06) in patients who completed 1-year treatment of pyrotinib. This is an interesting and uplifting results, given that in ExteNET study, the 2-year iDFS rate was 93.9% (95% CI 92·4-95·2) in the 1-year neratinib group, and the 5-year iDFS survival was 90.2%, and 1-year treatment of neratinib in ExteNET study did not translate into OS benefit after 8-year follow-up. In this case, readers will be eagerly anticipating the long-term follow-up results of the current PERSIST study, as well as the results of the phase III clinical trial (NCT03980054).I have the following comments:(1) The introduction of the differences between pyrotinib and neratinib in terms of mechanism, efficacy, resistance, etc. is supposed to be included in the text so that authors could better highlight the clinical significance of the current trial.

Thanks for this comment.

In terms of mechanism, pyrotinib and neratinib are both irreversible pan-HER tyrosine kinase inhibitors that target HER1, HER2 and HER4 by covalently binding to ATP binding sites. Overall, the similarities between them far outweigh the differences. This is the reason why we referenced the ExteNET study, which used neratinib as extended adjuvant therapy, for the sample size calculation.

Regarding efficacy, currently, no head-to-head studies comparing efficacy of pyrotinib and neratinib have been reported, and the comparison of the efficacy between them using historical data from different studies have inevitable bias due to differences in treatment regimens, study populations, assessment criteria, etc.

Regarding resistance, only a few studies with small sample size and case reports have investigated their mechanisms of resistance, and the underlying mechanisms have not been fully understood.

Collectively, we believe that the similarities in the mechanisms of these two drugs far outweigh their differences, and their efficacy and resistance cannot be reasonably compared. Moreover, the sample size calculation was conducted based on the premise that the two drugs are similar. After careful consideration, we believe that overanalyzing the differences between neratinib and pyrotinib would shift the focus of this manuscript. Therefore, we did not discuss their differences in the article.

(2) Please make sure that a total of 141 patients were enrolled in the study, 38 patients had a treatment duration of less than or equal to 6 months, and a total of 92 and 31 patients completed 1-year and 6-month treatment of extended adjuvant pyrotinib, respectively, which means 7 patients had a treatment duration of fewer than 6 months.

Thank you for raising this relevant question. There were 141 patients enrolled in the study and received study treatment, and a total of 92 and 31 patients completed 1-year and 6-month treatment of extended adjuvant pyrotinib. Of the remaining 18 patients, 16 patients had a treatment duration of fewer than 6 months, and 2 patients had a treatment duration longer than 6 months but less than 1 year.

(3) The previous surgery history should be provided, and how many patients received lumpectomy, and mastectomy.

Thank you for your suggestion. All patients in the present study underwent breast cancer surgery. Unfortunately, we did not collect data on the specific types of surgeries performed.

**Recommendations for the authors:**

**Reviewing Editor:**
I have carefully reviewed the content and findings of your study, and while I recognize the potential impact of your research, there are several critical aspects that need to be addressed to fully appreciate the contribution of your work.Significance of Findings:Your study provides valuable insights into the efficacy and safety of pyrotinib as an extended adjuvant therapy following trastuzumab-based treatment in patients with high-risk HER2-positive breast cancer. The 2-year invasive disease-free survival (iDFS) rate of 94.59% is notably high and suggests that pyrotinib could be a promising option for patients who have completed trastuzumab therapy. This is particularly significant given the unmet need for effective therapies that can extend disease-free survival in this patient population.Strength of Evidence:The strength of the evidence presented is supported by the multicenter phase II trial design, which included a substantial number of patients across 23 centers in China. The rigorous methodology, including the use of the Kaplan-Meier method for estimating iDFS and the application of the Brookmeyer-Crowley method for confidence intervals, adds to the credibility of your findings. However, the single-arm study design without a control group limits the ability to draw definitive conclusions about the comparative effectiveness of pyrotinib.In conclusion, your study presents intriguing findings that contribute to the field of breast cancer therapy. However, the current evidence, while suggestive of pyrotinib's potential, requires further validation in controlled trials to confirm its efficacy and optimal use in clinical practice. I encourage you to address the issues raised and consider resubmitting a revised version of your work.

Thank you for your comments. We acknowledge the limitation of our single-arm study design without a control group and agree that it restricts definitive conclusions about the comparative effectiveness of pyrotinib. This limitation was noted in our manuscript. Furthermore, we have revised our manuscript in response to the issues raised by the reviewers.